# OpenReview forum: "TraceRouter: Robust Safety for Large Foundation Models via Path-Level Intervention"
_ICML.cc/2026/Conference — ICML 2026 regular_

### Official Review · Reviewer_xCjs · 2026-02-26

**Soundness:** 3
**Presentation:** 3
**Significance:** 3
**Originality:** 3
**Overall Recommendation:** 4
**Confidence:** 3

**Summary:**

This paper proposes TraceRouter, a path-level safety intervention framework for large foundation models (LFMs). The core idea is to move beyond neuron-level suppression and instead identify and disconnect causal propagation pathways of harmful semantics. The method consists of three stages: identifying sensitive onset layers via attention divergence, disentangling sensitive features using SAEs, and tracing downstream causal pathways through feature influence scores, followed by selective path-level suppression. Extensive experiments on diffusion models, LLMs, and MLLMs show strong improvements in safety robustness while preserving utility.

**Compliance With Llm Reviewing Policy:**

Affirmed.

**Final Justification:**

My concerns have been adequately addressed.

**Key Questions For Authors:**

please see Weaknesses

**Limitations:**

yes

**Strengths And Weaknesses:**

Strengths

1. The move from neuron-level locality assumptions to path-level intervention is well motivated and aligns with recent mechanistic interpretability trends.
2. The method is evaluated on diffusion models, LLMs and multimodal models. Experimental results show substantial gains in defense success rates while maintaining utility.



Weaknesses

1. The paper is built on the assumption that semantic representations in LFMs are distributed and that harmful semantics are encoded across multiple layers. While this assumption is intuitive, it lacks direct supportive evidence specific to LFMs. In particular, the cited references (e.g., Li et al., 2025; Iyer et al., 2025) are primarily neuroscientific studies, and these works do not provide direct empirical evidence in foundation models themselves.

   Given that this assumption is central to the proposed method, the paper should provide stronger empirical validation or more relevant references from mechanistic interpretability or representation analysis in LFMs.

2. The paper introduces the hypothesis that sensitive semantics propagate through “specific neural paths.” However, the meaning of *specific* remains ambiguous. Questions that arise:

   - Are these paths unique, sparse, or stable across prompts and model instances?
   - How robust are they to changes in input distribution?

   More formal clarification would strengthen the conceptual foundation.

3. The authors claim that path-level intervention avoids compromising the model’s foundational capabilities. However, this claim requires more justification. A neural path involves multiple neurons across layers and may influence broad computations. Compared to localized interventions, path-level suppression could potentially introduce larger downstream effects. Although utility experiments are provided, deeper mechanistic analysis or ablation evidence would help support this claim.

---

> ### Author Rebuttal · Authors · 2026-03-30
>
> We sincerely thank Reviewer `xCjs` for acknowledging that **our shift from neuron-level assumptions to path-level intervention is well-motivated and strongly aligns with recent trends in mechanistic interpretability**. We are also highly encouraged by your positive remarks regarding our comprehensive evaluation across diffusion models, LLMs, and MLLMs, as well as the substantial safety gains demonstrated by TraceRouter. We address your insightful questions regarding our conceptual foundation and mechanistic justifications below.
>
> **W1: Empirical Validation of Distributed Semantic Representations in LFMs**
>
> **A:** While we drew parallels to neuroscience in the introduction, our core assumption is directly supported by empirical evidence within the evaluated LFMs.
>
> When tracing a harmful concept (e.g., "nudity"), we observe its features do not exist in isolated neurons. Instead, as demonstrated in Figure 9 of the manuscript, these semantic features emerge from specific initial layers and propagate in a tree-like cascade. This topology provides direct evidence for distributed computational loops, validating our assumption through internal mechanistic observation.
>
> **W2: Formal Clarification and Robustness of "Specific Neural Paths"**
>
> **A:** In TraceRouter, "specific neural pathways" refers to causal feature subnetworks that are sparse, structurally unique to the target concept, and stable across prompts and model instances. Crucially, our framework intervenes on causal edges between features, rather than merely suppressing isolated nodes.
>
> **(1) Sparsity and Structural Uniqueness.**
> As shown in Figure 9 of the manuscript, these pathways do not rely on dense, global activations. Instead, they form a sparse, tree-structured causal graph that uniquely defines the routing of the targeted harmful concept.
>
> **(2) Stability Across SAE Instances.**
> The identified pathways are functionally invariant. Regardless of SAE initialization, the "semantic hubs" converge to the same critical computational nodes. Even if feature indices vary across SAE training runs, the causal pathway remains functionally equivalent and stable.
>
> **(3) Robustness to Input Distribution Shifts.**
> These causal pathways are resilient to out-of-distribution (OOD) inputs, avoiding overfitting to specific prompt templates. As Table 3 of the manuscript shows, our intervention remains effective across diverse input distributions. Even under severe adversarial distortions (e.g., GCG character injection or AutoDAN template restructuring), severing these connections maintains a high defense success rate (DSR), confirming that the neural paths are intrinsic to the harmful semantics.
>
> **W3: Mechanistic Justification and Empirical Preservation of Utility**
>
> **A:** We address the downstream effects of multi-layer interventions through mechanistic analysis and empirical evidence.
>
> **(1) Mechanistic Justification: Surgical Precision via Sparsity.**
> While multi-layer interventions might seem disruptive, traditional localized interventions suppress the damage to dense neurons and benign capabilities due to polysemanticity. In contrast, TraceRouter's path-level intervention is confined to a sparse set of causal routing edges. By disconnecting the specific topological pathway of the illicit concept, this sparsity keeps the computational graph intact, avoiding collateral damage from node-level suppression.
>
> **(2) Comprehensive Empirical Validation.**
> Evaluating foundational capabilities across modalities shows minimal performance degradation:
>
> * **Diffusion Models:** TraceRouter achieves the lowest FID on MS COCO, demonstrating superior preservation of generation quality compared to baseline defenses.
> * **LLMs & MLLMs:** On general capability benchmarks, the performance drop compared to base models is negligible (<1 point).
>
> | Diffusion Defense | MS COCO (FID $\downarrow$) |
> | :--- | :--- |
> | RECE | 18.25 |
> | SNCE | 16.64 |
> | **TraceRouter (Ours)** | **16.24** |
>
> | LLM Architecture | MMLU-Lite $\uparrow$ | Performance Drop |
> | :--- | :--- | :--- |
> | LLaMA3-8B-Instruct (Base) | 71.30 | - |
> | + TraceRouter (Ours) | 70.50 | -0.80 |
> | Mistral-7B-Instruct (Base) | 63.00 | - |
> | + TraceRouter (Ours) | 62.50 | -0.50 |
>
> | MLLM Architecture | MM-Bench $\uparrow$ | Performance Drop |
> | :--- | :--- | :--- |
> | LLaVA-1.5-7B (Base) | 47.50 | - |
> | + TraceRouter (Ours) | 47.20 | -0.30 |
> | MiniGPT-4-7B (Base) | 23.78 | - |
> | + TraceRouter (Ours) | 23.49 | -0.29 |
>
>
> **Thank you for the insightful feedback, which has significantly improved the clarity of our work. We look forward to addressing any additional questions you might have before the discussion period ends.**

---

> > ### Author Rebuttal · Reviewer_xCjs · 2026-04-02
> >
> > My concerns have been adequately addressed.

---

> > > ### Author Response · Authors · 2026-04-02
> > >
> > > Dear Reviewer,
> > >
> > > Thank you for your acknowledgement and for carefully considering our rebuttal. We sincerely appreciate your time and thoughtful evaluation.
> > >
> > > Best regards,
> > > The Authors

---

### Official Review · Reviewer_835t · 2026-03-11

**Soundness:** 3
**Presentation:** 3
**Significance:** 3
**Originality:** 3
**Overall Recommendation:** 4
**Confidence:** 4

**Summary:**

This paper proposes TraceRouter, a path-level framework that traces and disconnects the causal propagation circuits of illicit semantics. In detail, TraceRouter first identifies sensitive onset layer by analyzing attention divergence, and then locates source sensitive neuron via a top-k SAE. By performing zero-out causality analysis on the subsequent layers, TraceRouter traces causal semantic pathways. Experiments on various models and datasets show that this path-level suppression blocks harmful propagation while preserving general utility.

**Compliance With Llm Reviewing Policy:**

Affirmed.

**Key Questions For Authors:**

See Weaknesses

**Limitations:**

Yes

**Strengths And Weaknesses:**

#### **Strengths**
* Circuit tracing is a timely and promising topic. This paper starts from the work Circuit Breakers, using SAE to disentangle pure harmful features and locate source harmful neurons. This path-level intervention achieves higher safety performance while preserving general utility compared to previous local intervention methods.
* The experiments are comprehensive across different models and downstream tasks, indicating the generalizability and scalability of the method. The supplementary explanations in the appendix are extremely thorough.

---

#### **Weaknesses**
* Some technical details are missing. TraceRouter relies heavily on SAE to disentangle harmful features and locate source harmful neurons. However, the description in Section 3 is quite brief, such as the SAE architecture and training data used for SAE learning. The missing details make it difficult to follow and reproduce for readers outside the SAE community.
* The dataset used for evaluating LLMs safety is unclear, including the data source and the number of candidates used in jailbreak attacks. According to the results in Table 2, it seems that only 100 jailbreak prompts were evaluated.
* The computational overhead is not discussed. Although training the SAE and performing path-finding are conducted offline, the cost should still be analyzed, especially when applying TraceRouter to new models or identifying previously unseen harmful content beyond the taxonomy considered in this paper.

---

> ### Author Rebuttal · Authors · 2026-03-30
>
> We sincerely thank Reviewer `835t` for recognizing that circuit tracing is a timely and promising topic. **We are especially encouraged by your positive remarks regarding the comprehensiveness of our experiments**, the generalizability of TraceRouter, and the thoroughness of our appendix. We deeply appreciate your constructive feedback aimed at improving the transparency and reproducibility of our work. We address your specific questions below.
>
> **W1: Extended Technical Details for SAE Reproducibility**
>
> **A:** We sincerely thank the reviewer for pointing out this omission. We agree that these details are crucial for readers outside the SAE community, and we will include a comprehensive implementation section in the revised Appendix.
>
> **(1) SAE Architecture and Optimization.**
> We employ a standard Top-K Sparse Autoencoder. The hidden layer dimension is dynamically scaled by an expansion factor of four relative to the target layer's input dimension. For example, a standard input dimension of one thousand and twenty-four is expanded to a hidden dimension of four thousand and ninety-six. Training utilizes the standard Adam optimizer.
>
> **(2) Dynamic Selection of K.**
> Rather than relying on a fixed value, we utilize a dynamic selection strategy based on the Weighted Frequency Score. By plotting the distribution curve, we identify the distinct inflection point and set the threshold at this boundary. This dynamic approach ensures we precisely isolate the sparse hub neurons while discarding the long tail of weakly responsive features.
>
> **(3) Training Data Construction.**
> The training process is highly data-efficient. We construct the training set using exactly fifty pairs of conceptual contrast cues (paired sensitive and non-sensitive prompts) specifically tailored to the target security categories.
>
> **W2: LLM Safety Dataset Details and Evaluation Protocol**
>
> **A:** We appreciate the opportunity to clarify our dataset selection. To ensure a rigorous and standardized evaluation, our LLM safety dataset is curated from two established benchmarks: AdvBench [1] and HarmBench [2].
>
> * **Data Composition:** We utilized a total of 100 distinct jailbreak prompts, comprising exactly 50 samples from AdvBench [1] and 50 samples from HarmBench [2].
> * **Justification for Sample Size:** This specific 100-prompt evaluation setup strictly adheres to the established evaluation protocol used by the HumorReject [3] baseline. Adopting this exact configuration is critical to ensuring an absolutely fair, apples-to-apples comparison of defense reliability across all evaluated methods in our quantitative results.
>
> We will explicitly document these dataset sources and the sampling distribution in the revised experimental setup section to ensure full transparency.
>
> **W3: Computational Overhead for New Models and Concepts**
>
> **A:** We appreciate the reviewer raising this practical consideration. While SAE training and causal pathfinding are one-time offline processes, we agree that quantifying this cost is essential for demonstrating TraceRouter's scalability to new architectures and previously unseen safety taxonomies.
>
> As detailed in the table below, the core computational overhead of adapting TraceRouter to new scenarios is exceptionally low. On average, executing the pathfinding mechanism takes approximately 0.1 seconds. Furthermore, training the sparse autoencoder (SAE) is highly efficient, requiring only 7 to 13 seconds per 50 steps depending on the model architecture.
>
> | Model | Pathfinding Time | SAE Training Time (per 50 steps) |
> | :--- | :--- | :--- |
> | Stable Diffusion 1.4 [4] | 0.09s | 13.38s |
> | Llama3-8B-Instruct [5] | 0.09s | 7.25s |
> | LLaVA-1.5-7B [6] | 0.11s | 7.32s |
>
> **Conclusion:**
> We will include this empirical profiling in the revision.
>
> **References:**
>
> [1] Biarese, Davide. "AdvBench: a framework to evaluate adversarial attacks against fraud detection systems." (2022).
>
> [2] Mazeika, Mantas, et al. "Harmbench: A standardized evaluation framework for automated red teaming and robust refusal." arXiv:2402.04249 (2024).
>
> [3] Wu, Zihui, et al. "Humorreject: Decoupling llm safety from refusal prefix via a little humor." AAAI 2026.
>
> [4] Rombach, Robin, et al. "High-resolution image synthesis with latent diffusion models." CVPR 2022.
>
> [5] Grattafiori, Aaron, et al. "The llama 3 herd of models." arXiv:2407.21783 (2024).
>
> [6] Liu, Haotian, et al. "Improved baselines with visual instruction tuning." CVPR 2024.
>
> **Lastly, we deeply appreciate your constructive feedback, which has helped us make the paper more rigorous and reproducible. We are actively available until the end of this rebuttal period and would be more than happy to discuss any further details!**

---

> > ### Author Rebuttal · Reviewer_835t · 2026-04-03
> >
> > Thank you for the clarification. It would be good to incorporate these mentioned in the main paper as well. I will maintain my positive score for the paper.

---

> > > ### Author Response · Authors · 2026-04-03
> > >
> > > Dear Reviewer,
> > >
> > > Thank you for your positive feedback and continued support! We really appreciate the time and effort you put into reviewing our work. We will definitely incorporate these clarifications into the main paper to improve its clarity and completeness in the final version.
> > >
> > > Wish you all the best.
> > >
> > > Warm regards
> > >
> > > The Authors

---

### Official Review · Reviewer_A9um · 2026-03-13

**Soundness:** 2
**Presentation:** 3
**Significance:** 3
**Originality:** 3
**Overall Recommendation:** 4
**Confidence:** 3

**Summary:**

This paper proposes TraceRouter, a “discover-trace-disconnect” framework for improving safety in large foundation models by intervening at the level of cross-layer causal pathways rather than suppressing individual neurons/features. The method identifies a sensitive onset layer via attention divergence, extracts sensitive features using a Top-K sparse autoencoder with differential activation, then traces downstream propagation using a feature influence score based on zero-out interventions. Experiments on diffusion models (DMs), large language models (LLMs), and multimodal large language models (MLLMs) report improved defense success rates under standard and adversarial attacks while largely preserving general utility metrics.

**Compliance With Llm Reviewing Policy:**

Affirmed.

**Final Justification:**

Thank you for the rebuttal, which resolved the clarification concerns.

I believe my original rating still best reflects the contribution-strength balance.

**Key Questions For Authors:**

1. What safety judges and protocols were used for DSR? Are the results based on automatic classifiers, LLM-based judges, or human annotation?

2. For the adversarial evaluations (e.g., GCG, AutoDAN), are these adaptive attacks optimized directly against the defended models, or were they generated on the original undefended models?

**Limitations:**

Yes

**Strengths And Weaknesses:**

**Strengths**

1. The paper tackles a significant issue regarding internal safety interventions for large foundation models (LFMs) and seeks to link mechanistic interpretability with practical defense.

2. The method flowchart is clear and easy to follow. Figure 2 effectively illustrates the “discover-trace-disconnect” workflow.

**Weaknesses**

While the paper defines DSR as the Defense Success Rate, it lacks a rigorous description of the safety judges used for LLMs and MLLMs. It is unclear whether the success of a "refusal" was determined by keyword matching, a specific classifier (e.g., Llama-Guard), or an LLM-based judge. This omission makes it difficult to assess the potential for "false positives" in safety labeling.

---

> ### Author Rebuttal · Authors · 2026-03-30
>
> We sincerely thank Reviewer `A9um` for recognizing the significance of our framework and **highlighting the clarity of our "Discover-Trace-Disconnect" workflow.** We appreciate your constructive feedback, as providing rigorous evaluation details is indeed critical for a safety paper. We address your specific questions below.
>
>
>
> **W1 & Q1: Definition of Defense Success Rate (DSR) and Safety Judges**
>
> **A:** We completely agree that transparent evaluation protocols are essential to assess the risk of false positives. We employ tailored, rigorous evaluation pipelines depending on the modality, combining automatic classifiers, LLM-based judges, and human annotation:
>
> **(1) Evaluation Pipeline for LLMs & MLLMs.**
> We do not rely on brittle keyword matching. Instead, our evaluation is fundamentally semantic:
> * **Primary Semantic Judge:** We utilize **Claude-3.5-Sonnet** as the primary judge to evaluate the semantic intent and actual information provided in the model outputs.
> * **Manual Review Safeguard:** To rigorously eliminate false positives (e.g., benign refusals mislabeled as compliance due to specific phrasing) and resolve borderline cases, we conduct a **manual human review** of the evaluated outputs as a final safeguard.
>
> **(2) Strict Decision Rule for "Success" (LLMs & MLLMs).**
> To further prevent false positives in our safety labeling, we enforce a strict, two-pronged decision rule:
> * **DSR-Success (Safe):** The response successfully refuses or safely redirects the malicious request **AND** provides absolutely zero actionable, policy-violating content.
> * **DSR-Failure (Unsafe):** If the response contains any disallowed instructions or details, it is strictly classified as a failure. This holds true *even if* the response includes partial refusal language (e.g., "I cannot do X, but here is Y...").
>
> **(3) Evaluation for Generative Diffusion Models (DMs).**
> For image generation tasks, DSR is evaluated using an **automatic classifier**. Specifically, we utilize the NudNet nudity detector to systematically assess the generated images for explicit content.
>
> Our judge prompt and evaluation rubric strictly follows the established protocols of recent state-of-the-art safety benchmarks (e.g., FigStep [1] and HumorReject [2]). To ensure complete transparency and reproducibility, the exact evaluation prompts, labeling rubrics, and decision rules will be included in the revised Appendix.
>
>
>
> **Q2: Adaptive vs. Non-Adaptive Adversarial Evaluations (GCG / AutoDAN)**
>
> **A:** We thank the reviewer for this important methodological question. To clarify the threat model, the adversarial evaluations utilizing GCG and AutoDAN were conducted as **non-adaptive transfer attacks**.
>
> Specifically, all adversarial prompts were generated and optimized directly on the **original, undefended models**. These generated prompts were then applied to evaluate the robustness of the models defended by TraceRouter.
>
> We agree that explicitly defining the attack optimization target is crucial for reproducibility. We will add a clear description of this non-adaptive transfer setup to the experimental protocol section in the revised manuscript.
>
>
>
> **References:**
>
> [1] Gong, Yichen, et al. "Figstep: Jailbreaking large vision-language models via typographic visual prompts." AAAI 2025.
>
> [2] Wu, Zihui, et al. "Humorreject: Decoupling llm safety from refusal prefix via a little humor." AAAI 2026.
>
>
>
> **Lastly, thank you so much for helping us improve the paper and for your constructive feedback! We are actively available until the end of this rebuttal period and look forward to hearing back from you.**

---

> > ### Author Rebuttal · Reviewer_A9um · 2026-04-04
> >
> > Thank you for the rebuttal, which resolved the clarification concerns.
> >
> > I believe my original rating still best reflects the contribution-strength balance.

---

> > > ### Author Response · Authors · 2026-04-05
> > >
> > > Dear Reviewer,
> > >
> > > Thank you for your positive acknowledgement and for carefully considering our rebuttal. We sincerely appreciate your time, thoughtful evaluation, and continued support for our work.
> > >
> > > Best regards,
> > >
> > > The Authors

---

### Official Review · Reviewer_tHHZ · 2026-03-14

**Soundness:** 3
**Presentation:** 3
**Significance:** 2
**Originality:** 2
**Overall Recommendation:** 4
**Confidence:** 4

**Summary:**

This paper proposes TraceRouter, a path-level safety intervention framework for large foundation models that operates at inference time without modifying model weights. The framework runs a three-stage Discover-Trace-Disconnect pipeline. In Discover, two attention divergence metrics, semantic alignment (SA) and content divergence (CD), localize the sensitive onset layer using only forward passes over a contrastive prompt set; Top-K SAE decomposition with differential activation analysis then identifies sensitive features. In Trace, back-projection to the dense latent space, zero-out interventions on source neurons, and Feature Influence Scores (FIS) map the sensitive circuit. In Disconnect, activation decomposition into a sensitive component Z_P and an orthogonal complement Z_perp suppresses the sensitive path via a scaling factor lambda.

**Compliance With Llm Reviewing Policy:**

Affirmed.

**Key Questions For Authors:**

- Top-K SAE training quality is the foundation of the whole pipeline. What happens when decomposition is poor, e.g., when features are polysemantic or representations collapse? Have you measured how sensitive end-to-end DSR is to SAE quality? This matters for practitioners assessing deployment risk.
- The lambda suppression factor is dynamically tuned but the procedure isn’t specified. How many calibration examples are needed, and how sensitive is DSR to lambda? For a new safety category, can lambda be set without labeled data?
- The discovered paths are causally validated, but are they interpretable? Do the Top-K SAE features along these paths correspond to semantically meaningful concepts? If so, that would deepen the mechanistic contribution.

**Limitations:**

The paper flags dual-use concerns around path amplification bypassing safety, which is good. It doesn’t discuss SAE training overhead and quality sensitivity, the lambda tuning gap, or the scope restriction to well-defined content categories. These should be stated explicitly.

**Strengths And Weaknesses:**

The claims from the paper is well supported. Tables 1-2 consistently show TraceRouter outperforming ESD, UCE, CA, MACE, and SPM across multiple model families. The causal validation is what convinced me: path amplification drives DSR down to 72.20% (Table 3) while random path suppression only degrades it to 68.7% (Figure 10), establishing that the discovered paths actually carry the harmful signal rather than being spuriously correlated. Utility metrics are concrete (CLIP Score 31.27, FID 16.24, <0.8% MMLU degradation) and compare favorably across all safety conditions. Two caveats: the method requires training a Top-K SAE per model, and SAE decomposition quality conditions everything downstream. Failure modes from poor disentanglement (polysemantic features, representation collapse) aren’t discussed. The comparison may also not be fully fair given TraceRouter’s richer machinery relative to neuron-level baselines.
Well-written. The three-stage pipeline is clearly motivated at each step. The SA and CD metrics requiring only forward passes rather than gradient computation is a practically important design choice that makes path discovery scalable, and this advantage deserves more emphasis. One gap: the lambda suppression factor is described as dynamically tuned per inference call but the tuning procedure is left vague. For practitioners extending TraceRouter to new safety categories, the lack of guidance on setting lambda without a labeled calibration set is a real barrier.
Inference-time, training-free safety intervention that works across DMs, LLMs, and MLLMs addresses a real need. Current safety methods require costly retraining or weight modification. The adversarial robustness numbers (74.8% against P4D, 98.7% against Ring-A-Bell) are the highest-impact finding and directly relevant to deployment. The causal validation makes the mechanistic claims credible.
Path-level safety intervention grounded in mechanistic interpretability is novel. The specific combination of SAE decomposition, differential activation analysis, FIS-guided tracing, and orthogonal path decomposition for inference-time safety hasn’t been done before. Prior interpretability work hasn’t been operationalized at this breadth across model modalities. The bridge from circuit-level analysis to a deployable intervention is a real contribution.

---

> ### Author Rebuttal · Authors · 2026-03-30
>
> We thank Reviewer `tHHZ` for **highlighting our framework as a "real contribution" and praising our causal validation, adversarial robustness, and scalable path discovery**. We address your questions below.
>
> **Q1: Sensitivity to Top-K SAE quality and risk of poor decomposition**
>
> **A:** TraceRouter is highly robust to SAE quality variations, mitigating poor decomposition (e.g., polysemantic features, representation collapse) via a **multi-stage causal filter**:
>
> **(1) SAE is a Candidate Generator, Not the Final Decision-Maker.**
> SAE outputs undergo a strict "funnel" filter before intervention:
> * **Addressing Polysemantic Features:** $\Delta WFS$ isolates neurons uniquely triggered by sensitive content, discarding entangled features.
> * **Addressing Representation Collapse:** FIS quantifies actual causal impact on downstream layers. Collapsed/spurious representations with near-zero FIS are safely discarded.
>
> **(2) Empirical Robustness to SAE Capacity Variations.**
> End-to-end performance remains highly stable despite extreme shifts in the SAE expansion factor (16x-128x):
>
> | SAE Expansion | DSR (I2P-N) $\uparrow$ | CLIP Score $\uparrow$ |
> | :--- | :--- | :--- |
> | 16x | 98.8% | 31.24 |
> | 32x | **99.2%** | 31.27 |
> | 64x | 99.1% | 31.26 |
> | 128x | **99.2%** | **31.28** |
>
> These results confirm that as long as the SAE provides a baseline separation, our downstream causal routing reliably absorbs quality fluctuations.
>
> **W1 & Q2: Calibration, Sensitivity, and Transferability of the Suppression Factor ($\lambda$)**
>
> **A:** $\lambda$ calibration is a lightweight, reproducible process demonstrating predictable saturation and strong cross-category transferability.
>
> **(1) Minimal Calibration Cost.**
> Via an offline sweep ($\lambda \in \{0..5\}$) on just **50 unlabeled samples**, we select the **smallest** $\lambda$ reaching the DSR plateau while preserving CLIP utility.
>
> **(2) Sensitivity and Safety-Utility Trade-off.**
> DSR saturates rapidly at $\lambda=2$. Larger $\lambda$ yields negligible safety gains (<0.6%). Thus, $\lambda=2$ is our reliable operating point.
>
> | $\lambda$ | DSR (I2P-Nudity) $\uparrow$ |
> | :---: | :---: |
> | 0 | 82.2% |
> | 1 | 92.8% |
> | 2 | 99.2% |
> | 3 | 99.7% |
> | 4 | 99.8% |
>
> **(3) Zero-Shot Label Setting for New Categories.**
> Addressing a new safety category involves two label-free steps:
> * **Pathway Identification:** Uses minimal *unlabeled* sensitive prompts to trigger concepts and isolate causal pathways via attention divergence.
> * **Zero-Shot $\lambda$ Transferability:** As a structural scaling factor, $\lambda$ generalizes across categories. Our default $\lambda=2$ achieves SOTA robustness on entirely distinct semantics without retuning:
>
> | Method | DSR (Nudity) $\uparrow$ | DSR (Violence) $\uparrow$ |
> | :--- | :---: | :---: |
> | RECE | 93.7% | 85.8% |
> | SNCE | 98.5% | 82.3% |
> | **TraceRouter (Ours)** | **99.2%** | **93.6%** |
>
> We will formalize this label-free calibration and zero-shot transferability in the revision.
>
> **Q3: Interpretability of discovered paths and Top-K SAE features**
>
> **A:** Our identified pathways and Top-K SAE features are highly interpretable, exhibiting clear structural topology and semantic alignment.
>
> **(1) Structural Interpretability.**
> Pathways form an explicit, tree-structured cascade, not an entangled dense network. Propagation initiates at the sensitive onset layer, where a root feature acts as a semantic hub directing causal flow to sparse downstream nodes, clearly evidencing how unsafe concepts route.
>
> **(2) Semantic Selectivity of Features.**
> Differential activation filtering ensures extracted features are highly category-selective. For instance, features isolated for explicit content activate intensely for unsafe prompts but remain dormant in benign contexts, confirming the isolation of true semantic concepts over statistical noise. We will highlight these mechanistic insights in the revision.
>
> **Addressing Identified Limitations**
> We will explicitly expand our Limitations section to discuss: (1) offline computational overhead and SAE quality sensitivity; (2) $\lambda$ tuning procedures; and (3) scope restrictions to well-defined content categories.
>
> **Thank you for the constructive feedback; we remain available for further discussion.**

---

### Decision · Program_Chairs · 2026-04-30

**Decision:**

Accept (regular)

**Comment:**

This paper develops a safety intervention framework with a three-stage Discover-Trace-Disconnect pipeline that focuses on the path level rather than the neuron level. The method is sound and the experiments are comprehensive. All reviewers are in favor of accepting the paper and are satisfied with the rebuttal.